

**The effects of different environmental factors on biochemical composition of particulate**
**organic matters in Gwangyang Bay, South Korea**
**Jang Han Lee[1], Dabin Lee[1], Jae Joong Kang[1], Hui Tae Joo[1], Jae Hyung Lee[1], Ho Won**
**Lee[1], So Hyun Ahn[1] and Sang Heon Lee[1]**
[1]Department of Oceanography, Pusan National University, Geumjeong-gu, Busan 46241,
Korea
*Corresponding author: sanglee@pusan.ac.kr



## Abstract

Biochemical composition of particulate organic matter (POM) through phytoplankton photosynthesis is important to determine food quality for planktonic consumers as well as physiological conditions of phytoplankton. Major environmental factors controlling for the biochemical composition were seasonally investigated in Gwangyang Bay which has only natural conditions (e.g., no artificial dams) in South Korea. Water samples for the biochemical compositions were obtained from three different light depths (100%, 30%, and 1%) mainly at 3 sites in Gwangyang bay from April 2012 to April 2013. Different biochemical classes (carbohydrates [CHO], proteins [PRT], and lipids [LIP]) were extracted and then the concentrations were determined by the optical density measured with a spectrophotometer. The highest and lowest of PRT compositions among the three biochemical classes were in April 2012 (58.0%) and August 2012 (21.2%), whereas the highest and lowest LIP compositions were in August 2012 (49.0%) and April 2012 (24.8%), respectively. CHO composition was recorded high in January 2013 and maintained above 25% during the study period. The calorific contents of food material (FM) ranged from 1.0 Kcal m$^{-3}$ to 6.1 Kcal m$^{-3}$ (annual mean ± S.D. = 2.8 Kcal m$^{-3}$ ± 1.1 Kcal m$^{-3}$). Based on Pearson's correlation coefficient analysis, a major governing factor for biochemical composition of POM was dissolved inorganic nitrogen loading from river-input in Gwangyang bay. In conclusion, relatively larger amount of FM and higher calorific contents of POM found in this study compared to other regions reflected good nutritive conditions for sustaining productive shellfish and fish populations in Gwangyang bay. Continuous observations are needed for monitoring marine ecosystem response to potential environmental perturbations in Gwangyang bay.

**Key words:**

Particulate organic matter, biochemical composition, phytoplankton, nitrogen source



## 1. Introduction

Particulate organic matter (POM) mostly from phytoplankton photosynthesis in the euphotic

layer is an important food source for planktonic consumers in water columns (Cauwet, 1978) and their
biochemical contents reaching the benthic environments are largely utilized by benthic organisms
(Nelson and Smith, 1986; Rice et al., 1994). Therefore, POM is an essential link between surface and
benthic ecosystems (Graf, 1992). Previous studies showed that the biochemical composition of the
POM such as protein (PRT), lipid (LIP) and carbohydrate (CHO) levels could provide useful
information on the nutritional value which is potentially available to consumers (Mayzaud et al, 1989;
Navarro et al., 1993; Navarro and Thompson, 1995). However, previous studies mainly focused on
the occurrence in the different patterns of biochemical composition of POM. It is noteworthy to
investigate how biochemical composition of POM responds to changes in various environmental
factors, such as nutrients, light, temperature, and salinity and to assess food quantity for higher trophic
levels.

The coastal areas represent one of the world's most vital aquatic resources, supporting and

providing food resources and habitats for large numbers of fish and shellfish species (Kwak et al.,
2012; Wetz and Yoskowitz, 2013; references therein). In Gwangyang bay, the southern coast of Korea
(Fig. 1), coastal fisheries and shellfish farming have been prevalence. Over the past decades, the bay
have become industrialized such as the construction of steel mill company, power plant and industrial
complex and environmental disturbances have been predicted. Also, estuaries have a high short-term
variability depending on many episodic events, such as freshwater inputs, tidal cycles (neap-spring),
and wind (storms) (Cloern and Nichols, 1985). These anthropogenic forces and environmental
changes drastically affect the estuarine habitat properties which can cause different biochemical
compositions of POM. Unfortunately, little information is yet available on the biochemical
composition of POM in the bay, South Korea. Hence, this study tested the question of the main
environmental factors determining seasonal variation and of biochemical composition POM and





assessed quantity of food material (FM) in the bay. Physical (temperature, salinity, irradiance, river-
input and rainfall data), chemical (nutrients), and biological (chlorophyll-*a* [chl-*a*], particulate organic
carbon [POC] and nitrogen [PON]) parameters were measured in order to both characterize the origin
of POM and understand their effects on the biochemical composition of POM. The aims of this study
were to: (1) investigate seasonal variation of biochemical composition of POM, (2) identify the origin
of POM, and (3) determine a major governing environmental factor for biochemical composition of
POM.
**2. Materials and methods**
**2.1. Study site and sampling procedure**

The study site was located in Gwangyang Bay (34.9 ° N, 127.8 ° E), the southern coast of

Korea (Fig. 1). The total area of the bay is 230 km$^2$ at mean sea level (Kang et al., 2003). The bay is
characterized by semidiurnal tides with a maximal range of about 4.8 m at spring tides (Korea
Hydrographic and Oceanographic Administration). Freshwater flows into the bay from the Seomjin
River at the northern part of the bay (mean flow 27 m$^3$ s$^{-1}$ and annually 1.9 x 10$^9$ t during the study
period; the National Institute of Environmental Research) and seawater enters through the narrow
southern channel (Yeosu Channel).

To obtain data for seasonal variation of POM in the euphotic depth, the field samplings were

undertaken at 3 stations of the bay (St.1 or St. 2A, St. 4, and St. 5; see Fig. 1) on a seasonal basis April,
June, August, and October in 2012 and January and April in 2013. St. 1 was changed to St. 2A after
April 2012 because of logistic problems. Both stations have similar environmental conditions at a
relatively close distance. Using a 5 L Niskin water sampler, water samples were collected at different
depths of 3 light intensities (100%, 30%, and 1% of surface irradiances; hereafter 3 light depths)
which were determined by a secchi disk and transferred to brown sample bottles which were
previously washed with a solution of 0.1 N HCl. To obtain *in situ* physical parameters, water



temperature and salinity were measured with YSI-30 (YSI incorporated) and photosynthetically active
radiation (PAR) was measured by a quantum sensor (LI-190SA, LI-COR) with a data logger (LI-1400,
LI-COR). Rainfall and river input data during the study period were obtained from the Korea
Meteorological Administration (http://www.kma.go.kr/index.jsp) and the National Institute of
Environmental Research (http://water.nier.go.kr/main/mainContent.do).
**2.2. Chl-*a* and major inorganic nutrient analysis**

In order to determine chl-*a* concentration, water samples from 3 light depths were filtered

through 25 mm GF/F (Whatman, 0.7 µm) which were kept frozen immediately and returned to the
laboratory at Pusan National University, Korea for a further analysis. The filters for chl-*a*
concentration were extracted in 90% acetone in a fridge (4 °C) for 24 h and centrifuged for 20
minutes at 4000 rpm. Using a fluorometer (Tuner Designs, 10-AU) which had been calibrated with
commercially purified chl-*a* preparations, chl-*a* concentrations were measured and calculated (Parsons
et al., 1984). Water samples for inorganic nutrient concentrations from surface and bottom waters
were obtained from Niskin bottles. The samples were kept frozen (-70 °C) and sent for analysis to the
laboratory in the East Sea Fisheries Research Institute (QUAATRO, Seal Analytical).
**2.3. Particulate organic carbon and nitrogen analysis**

The water samples were filtered through pre-combusted (450 °C) 25 mm GF/F (Whatman,

0.7 µm). The filters for POC, PON, and $\delta^{13}$C values were preserved frozen (-20 °C) and determined
using a Finnigan Delta + XP mass spectrometer at the stable isotope laboratory of the University of
Alaska Fairbanks, USA.
**2.4. Biochemical composition analysis**

The water samples for the biochemical composition (carbohydrates, proteins, and lipids) of

POM were collected from 3 light depths. The water samples were filtered through 47 mm GF/F



(Whatman, 0.7 µm pore), which were immediately frozen at -70 °C and preserved for biochemical
composition analysis at the home laboratory.

*Protein analysis*

Protein (PRT) concentrations were assessed according to a modified method of Lowry at el.
(1951). The filters for PRT analysis were transferred into 12 mL centrifuge tubes with 1 mL $DH_2O$,
respectively. The filters were grounded (using a glass rod) in the tubes with a 5 ml alkaline copper
solution (a mixture of 2% $Na_2CO_3$ in 0.1 N NaOH with 0.5% $CuSO_4 \cdot 5H_2O$ in 1 % sodium or
potassium tartrate; 50:1, v/v). The solutions for PRT concentrations were mixed well (using a vortex)
and allowed to stand for 10 min at room temperature in the hood. After 10 min, 0.5 mL of diluted
Folin-Ciocalteu phenol reagent (1:1, v/v) was added into the solution, mixed occasionally with a
vortex mixer, and allowed to sit for 1 h 30 min. The solutions with a blue color were centrifuged at
3,000 rpm for 10 min. Absorbance of the supernatant was measured at 750 nm. Bovine Serum
Albumin (2 mg mL$^{-1}$, SIGMA) was used as a standard for the PRT concentration.

*Lipid analysis*

Lipid (LIP) concentrations were extracted according to a column method modified from
Bligh and Dyer (1959), and Marsh and Weinstein (1966). The filters for LIP analysis were transferred
into 16 mL glass tubes with 3 mL of chloroform-methanol (1:2, v/v). The filters in the tubes were
grounded, and then the mixtures were mixed using a vortex mixer. For LIP extraction, glass tubes
with samples were stored in the fridge (4 °C) to prevent the solvents from evaporating. After 1 h, the
solvents were centrifuged at 2,000 rpm for 10 min and the supernatants were collected and stored in
new tubes. This extraction procedure was performed once again immediately. When the extractions
were completed, 4 mL of $DH_2O$ was added to the solution in the new tubes, and the solution was
homogenized using a vortex mixer. After mixing, the tubes were centrifuged at 2,000 rpm for 10 min,
and the solvents were separated into two phases (the chloroform phase for lipids and methanol +



DH$_2$O phase). The methanol + DH$_2$O phase was removed from the solvent using a Pasteur pipette.
The chloroform phase was placed in a dry oven at 40 °C for 48 h. After it totally dried for
carbonization analysis (Marsh and Weinstein 1966), 2 mL of H$_2$SO$_4$ was added to the tubes and they
were placed in a heating block at 200 °C for 15 min. After this heating procedure, the tubes were
quickly placed in a water bath at room temperature; 3 ml of DH$_2$O was added to the tubes and the
solvents were homogenized (with a vortex mixer) and stood for 10 min or until all bubbles had
disappeared. Absorbance of the supernatant was measured at 375 nm. Tripalmitin solutions were used
as a standard for the LIP concentration.
*Carbohydrate analysis*

Carbohydrate (CHO) concentrations were measured according to Dubois et al. (1956). The

POM samples for carbohydrate analysis were transferred individually into 15 mL polypropylene (PP)
tubes. After 1 mL of DH$_2$O was added to the PP tubes, the samples were grounded using a glass rod.
One ml of 5 % phenol for CHO extraction was added additionally, and the solutions were allowed to
stand for 40 min at room temperature in the hood. After the extraction, 5 mL of sulfuric acid (H$_2$SO$_4$)
was added to the solutions, mixed using a vortex mixer, and allowed to stand for 10 min. The
solutions with an orange-yellow color were centrifuged at 3,500 rpm for 10 min. Absorbance of the
supernatant was measured at 490 nm using UV spectrophotometer (Labomed, Germany). D (+) -
glucose solutions (1 mg mL$^{-1}$, SIGMA) were used as a standard for the CHO concentration.
**2.5. Statistical analyses and calorific value calculation**

Statistical tests were carried out using the statistic software "SPSS" (*t*-test, ANOVA and

Pearson's Correlation Coefficient). The level of significance was set at $p < 0.05$. The calorific value
(Kcal g$^{-1}$) of the food material (FM) (FM was defined by Danovaro et al. (2000); PRT + LIP + CHO
concentrations; hereafter FM) and the calorific content of FM (Kcal m$^{-3}$ = Kcal g$^{-1}$ × g FM m$^{-3}$) were
calculated using the Winberg (1971) equation.





**3. Results**
**3.1. Seasonal distribution and variation of environmental factors and chl-*a* concentrations**
The values of environmental factors were summarized in Table 1. The temperature ranged
from 5.5 to 26.1 °C. The temperature increased from April to August (the highest temperature in
August 2012 at St. 4: 26.1 °C) and decreased from August to January (the lowest temperature in
January 2013 at St. 2A: 5.5 °C). The salinity ranged from 14.5 to 32.9 ‰. Generally, the salinity
increased from St. 1 or St. 2A to St. 5. Relatively lower salinity, which is mainly affected by fresh
water input from the Seomjin River, was observed at St. 2A. The annual average euphotic depth was
6.5 ± 3.4 m, ranging from 2 to 12 m.
The highest nutrient concentrations were measured in April 2012, when the concentrations of
$NO_2 + NO_3$, $SiO_2$, $NH_4$, and $PO_4$ were above 5.0 µM, 2.0 µM, and 0.2 µM, respectively, except at 1%
light depth at St. 4. All inorganic nutrients except $SiO_2$ were nearly depleted in August 2012 (Table 1).
During the rest of our study period, $NO_2 + NO_3$ and $SiO_2$ concentrations were observed with similar
decreasing patterns from St.1 or St. 2A to St. 5. $NH_4$ concentrations averaged from October 2012 to
April 2013 were 1.1 µM ± 0.4 µM, ranging from 0.5 µM to 1.9 µM. $PO_4$ concentrations (average ±
S.D. = 0.1 ± 0.1 µM) ranged from 0 to 0.4 µM throughout the water columns at all stations except at
St. 2A in April 2012 during the study period.
Monthly rainfall and river-input in the study location ranged from 15.6 mm to 559.0 mm
(annual mean ± S.D. = 151.0 mm ± 155.5 mm) and 42.3 to 447.2 x $10^6$ t (annual mean = 144.4 x $10^6$ t),
respectively. Rainfall and river-input were recorded as high during summer and low during winter
(Table 2). Average irradiance during our incubation hour ranged from 167.9 ± 133.5 to 1593.3 ± 414.5
µmols m$^{-2}$ s$^{-1}$ (average ± S.D.) from April 2012 to April 2013. The highest and lowest irradiance were
recorded in April 2013 and April 2012, respectively**.**
Chl-*a* concentrations in the euphotic depth ranged from 0.8 µg L$^{-1}$ to 14.2 µg L$^{-1}$ during the




study period (annual mean ± S.D. = 3.4 μg L$^{-1}$ ± 2.8 μg L$^{-1}$; Table 1). There were no significant

differences of chl-$a$ concentrations among 3 light depths and spatial distribution. However, there was

seasonal variation of chl-a concentrations during study period. Chl-$a$ concentrations were increased

from April to August and decreased from August to October in 2012 and increased slightly again in

January and April 2013.

### 3.2. $\delta^{13}$C values and carbon to nitrogen ratios of POM

$\delta^{13}$C values of sea surface POM ranged from - 23.1 ‰ to - 16.5 ‰ and the annual average

$\delta^{13}$C value was -20.9 ‰ (S.D. = ± 3.2 ‰). The annual average carbon to nitrogen (C:N) ratio of POM

was 7.0 ± 0.4 (average ± S.D.), ranging from 6.8 to 7.7 (Table 3).

### 3.3. Seasonal variation of biochemical composition

The contents of CHO, PRT, and LIP of POM in the water column ranged from 14.2 μg L$^{-1}$ to

412.3 μg L$^{-1}$ (129.5 ± 87.2 μgL$^{-1}$), from 22.8 μg L$^{-1}$ to 382.4 μg L$^{-1}$ (155.0 ± 73.3 μgL$^{-1}$), and from

21.4 μg L$^{-1}$ to 401.4 μg L$^{-1}$ (154.9 ± 78.9μgL$^{-1}$), respectively (Table 4). The FM contents of POM

ranged from 170.9 μg L$^{-1}$ to 915.7 μg L$^{-1}$ (435.5 ± 175.5 μgL$^{-1}$). Since there were no significant

differences in biochemical concentrations of POM and FM among 3 light depths and spatial

distributions, we averaged each biochemical compound and FM on monthly basis. The CHO and LIP

concentrations increased from April to August and decreased from August to October in 2012. In

contrast, the PRT concentrations decreased from April to October in 2012 and increased from October

in 2012 to April in 2013. The seasonal pattern of FM concentrations was similar to the pattern of chl-$a$

concentrations.

In order to estimate the biochemical composition as food quality, we obtained relative

contributions of each biochemical concentration of POM to FM, based on percentage basis. The

biochemical composition of each class (CHO, PRT and LIP) ranged from 8.3% to 59.1%, from 6.8%

to 74.9% and from 9.4% to 68.3%, respectively (annual mean ± S.D. of CHO, PRT, and LIP





composition = 26.4 ± 9.4%, 37.8 ± 16.1%, and 35.7 ± 13.9%, respectively; Table 4). We found the
seasonal variation of biochemical composition based on monthly basis of biochemical composition
(Fig. 2). To illustrate these variations of biochemical composition of POM, the highest and lowest
PRT compositions were in April 2012 and August 2012. In contrast to PRT compositions, the highest
and lowest LIP compositions were in August 2012 and April 2012. CHO composition was recorded
high in January 2013, but to compare CHO composition to PRT and LIP composition, CHO
composition was not strong varied during the study period.
**3.4. Seasonal variations of the calorific values and contents of FM**

The calorific values and contents of FM ranged from 5.4 Kcal $g^{-1}$ to 7.9 Kcal $g^{-1}$ (annual

mean ± S.D. = 6.6 Kcal $g^{-1}$ ± 0.6 Kcal $g^{-1}$) and 1.0 Kcal $m^{-3}$ to 6.1 Kcal $m^{-3}$ (annual mean ± S.D. = 2.8
Kcal $m^{-3}$ ± 1.1 Kcal $m^{-3}$), respectively (Table 4). The calorific values of FM had no apparent seasonal
pattern, whereas the calorific contents of FM had a seasonal pattern similar to the seasonal variation
of FM concentrations.
**3.5. Relationship between biochemical pools and environmental conditions**

Relationships between biochemical pools and environmental conditions were performed

using Pearson's correlation matrix. Based on the results, we found a significant, positive relationships
between PRT composition and river-input (r = 0.84, $p < 0.01$, Table 5, Fig. 3) and PRT composition
and dissolved nitrogen concentrations ($NH_4$ : r = 0.69, $p < 0.01$; $NO_2+NO_3$ : r = 0.54, $p < 0.01$, Table
5). Lipid composition had an inverse relationships with river-input (r = -0.63, $p < 0.01$) and dissolved
nitrogen concentrations ($NH_4$ : r = -0.59, $p < 0.01$; $NO_2+NO_3$ : r = -0.53, $p < 0.01$). These
relationships led to a significant reverse relationship between PRT composition and LIP composition
(r = -0.81, $p < 0.01$, Fig. 4). PRT composition was negatively correlated with temperature (r = -0.52, $p$
< 0.01), whereas LIP composition was positively correlated with temperature (r = 0.72, $p < 0.01$).
There were no significant relationships between PRT composition and irradiance and LIP composition





and irradiance.
**4. Discussion**
**4. 1. Environmental conditions and chl–*a* concentration**

The annual average chl-*a* concentration during the research period was 3.4 µg L$^{-1}$ (S.D.=

±2.8 µg L$^{-1}$) with a range from 0.8 to 14.2 µg L$^{-1}$ which is in a similar range of chl-*a* concentrations
reported previously in Gwangyang bay, although it varied across different seasons and sampling
depths (Cho et al., 1994; Choi et al., 1998; Lee et al., 2001; Kwon et al., 2001; Jang et al., 2005; Yang
et al., 2005; Beak et al., 2011; Min et al., 2011; Beak et al., 2015). Previous studies reported that chl-*a*
concentration was influenced mainly by salinity, temperature, and nutrients (nitrate and phosphate)
depending on freshwater input from the Seomjin River. Our results in this study were similar to
former studies (r = 0.34 and -0.41, *p* < 0.05, n = 48 and 28 for salinity and NH$_4$, respectively).
However, high chl-*a* concentrations were previously recorded in spring and fall, whereas the highest
concentrations were observed in summer (August 2012) from this study. In fact, Baek et al. (2015)
reported that high chl-*a* concentrations were found in summer similarly, although there was difference
between environmental factors and chl-*a* concentrations as compared with our results. The high levels
of chl-*a* were observed with high nutrient concentrations and low salinity levels in the surface water
by Baek et al. (2015), whereas the high values existed with low nutrient concentrations and high
salinity levels in our results.

Despite this dissimilarity of environmental factors with high chl-*a* concentrations, we also

found the highest chl-*a* concentrations observed in summer. According to Shaha and Cho (2009),
there is a tendency with increasing precipitation and river-input in Gwangyang Bay during summer.
This trend could increase loading nutrients from freshwater for maintaining phytoplankton growth in
summer. In addition, a strong light intensity during summer could be favorable for phytoplankton
growth since our study area was extremely turbid conditions during almost all seasons due to




freshwater discharge and a strong spring-neap tidal oscillation. As a result, the combination of these
factors is believed to enhance chl-*a* concentration and primary production of phytoplankton during
summer in Gwangyang Bay.
**4. 2. POM characterization**

In general, POM consists of a mixture of living as well as detritus materials (phytoplankton,

bacteria, zooplankton, fecal pellets, terrestrial matters, etc.) originating from freshwater and estuarine
and marine environments. POM samples can be characterized or determined for source of the major
contributor(s). The C:N ratio generally ranges between 6 and 10 for phytoplankton, whereas the ratios
are between 3 and 6 for zooplankton and bacteria (Savoye et al, 2003; references therein). For
terrestrial organic matters, the C:N ratios are normally over 12 (Savoye et al, 2003; references therein).
Therefore, it is useful to classify phytoplankton from heterotrophs and terrestrial materials (Lobbes et
al., 2000; Savoye et al., 2003; Lee and Whitledge, 2005). In this study, the mean C:N ratios of POM
was 7.0 (S.D. = ± 0.4), which indicates that this POM is mainly phytoplankton (Table 3). However,
the C:N ratio must be used with caution because of its variation in the process of POM degradation
(Savoye et al, 2003). For example, PON is preferentially degraded compared to POC of
phytoplankton, which causes an increase of the C:N ratio. Terrestrial organic matters (high C:N ratio)
colonized by bacteria (low C:N ratio) lowers their initial high C:N ratio (Savoye et al, 2003;
references therein). Therefore, similar C:N ratios of POM could be produced by degraded
phytoplankton and bacteria-colonized terrestrial organic matters (Lancelot and Billen 1985; Savoye et
al, 2003). In addition to C:N ratios, $\delta^{13}C$ of POM can be used for determining their origin. Kang et al.
(2003) reported that the mean $\delta^{13}C$ signature of phytoplankton in Gwangyang Bay was -20.8 ‰ (S.D.
= ± 1.1‰). In this study, our mean $\delta^{13}C$ signature of POM was -20.9 ‰ (S.D. = ± 3.2‰), which also
indicates that POM was mostly phytoplankton during the study periods (Table 3). Based on our C:N
ratio and $\delta^{13}C$ value in this study, we confirmed that our POM samples were primarily comprised of
phytoplankton in Gwangyang Bay.



### 4. 3. Environmental conditions and biochemical pools

Biochemical pools of POM originating from phytoplankton are influenced by various environmental factors, such as temperature, salinity, nutrients, and light conditions (Morris et al., 1974; Smith and Morris, 1980; Rivkin and Voytek, 1987; Boëchat and Giani, 2008; Cuhel and Lean, 1987; Mock and Kroon, 2002; Khotimchenko and Yakoleva, 2005; Ventura et al., 2008; Sterner et al. 1997). In this study, significant relationships were found between environmental conditions and biochemical pools, especially PRT and LIP (Table 5). Temperature was positively and negatively correlated with LIP and PRT. Previous studies reported that higher temperature stress mainly affects nitrogen metabolism (Kakinuma et al., 2006) which is related to significant decrease of PRT with increases of LIP and CHO content (Tomaselli et al., 1988; Oliveira et al., 1999). In a high temperature-stressed condition of phytoplankton, the decrease in PRT content is related to breakdown of protein structure and interference with enzyme regulators (Pirt, 1975), whereas LIP is predominant because LIP is more closely associated with cell structure such as thickened cell walls (Smith et al., 1989; Kakinuma et al., 2001, 2006). Our results are in agreement with other works, as described above.

The relationships between nutrients and biochemical pools could be explained by nutrient limitation and the characteristics of each biochemical compound. A combination of nutrient concentrations and ratios can be used to assess nutrient limitation (Dortch and Whitledge, 1992; Justić et al., 1995). Dortch and Whitledge (1992) suggested that nutrient limitations are existed in the Mississippi river plume and Gulf of Mexico, if the dissolved inorganic phosphorus (DIP), dissolved inorganic nitrogen (DIN), and dissolved silicon (DSi) concentrations in water column are less than 0.2, 1.0 and 2.0 µM, respectively. In addition, molar ratios of the DIN:DIP and DSi:DIN can be indicators of nutritional status and the physiological behavior of phytoplankton (Redfield et al., 1963; Goldman et al., 1979; Elrifi and Turpin, 1985; Roelke et al. 1999). The following criteria of their molar ratios were (a) DSi:P ratio >16, and DSi:N ratio >16 for phosphorus (P) limitation; (b) DSi:DIP ratio >16 and DSi:N ratio >1 for nitrogen (N) limitation; (c) DSi:DIP ratio <16 and DSi:DIN ratio <1 for





silicate (Si) limitation. In this study, nutrient limitation conditions were observed by absolute nutrient
concentrations or/and their molar ratios depending on seasons (Table 6). Previous studies of
biochemical composition in relation to nutrient limitation reported that PRT production of
phytoplankton was enhanced under abundant N conditions (Fabiano et al., 1993; Lee et al., 2009). In
contrast, LIP production and storage were dominant (Shifrin and Chisholm, 1981; Harrison et al.,
1990) and PRT contents decreased (Kilham et al., 1997; Lynn et al., 2000; Heraud at al., 2005) under
N-depleted conditions. High LIP contents have also been detected in phytoplankton under P or/and Si
limitation (Lombardi and Wangersky, 1991; Lynn et al. 2000; Heraud et al., 2005; Sigee et al., 2007).
Under N or P-limited conditions, triglyceride content (energy storage) increases and shifts from PRT
to LIP metabolism since proteins are nitrogenous compounds whereas LIP and CHO are non-
nitrogenous substrates (Lombardi and Wangersky, 1991; Smith et al., 1997; Takagi et al., 2000). In
our study, Si and P concentrations may not significantly impact on biochemical composition of
phytoplankton. Si concentrations were almost above 2.0 µM except in April 2013 during the study
period. P limitation was observed based on the absolute concentration and molar ratios during study
period. However, under P limitation, phytoplankton can relocate the cellular P pool to maintain their P
requirements for the maximum growth rate (Cembella et al., 1984; Ji and Sherrell, 2008). In this
respect, we suggest that DIN could be significantly impact on biochemical composition of
phytoplankton in our study area. DIN was initially believed to be the most important limiting factor
for phytoplankton growth in marine ecosystems (Ryther and Dunstan, 1971; Howarth, 1988). In fact,
DIN was strongly positively correlated with PRT composition, whereas it was negatively correlated
with LIP composition. The most of DIN loading came from freshwater input of the Seomjin River
(Table 5, river-input vs $NH_4$ and $NO_2+NO_3$; r = 0.91 and 0.55, p < 0.01, respectively) influences on
PRT and LIP synthesis and subsequently macromolecular composition of phytoplankton. As a result,
the amount of river-input was also strongly correlated with PRT composition (Table 5 and Fig. 3).
Therefore, DIN is an important controlling factor for biochemical composition, especially PRT and
LIP composition of phytoplankton in Gwangyang bay.





Although irradiance is also known for an important governing factor for biochemical
composition, irradiance was not significantly correlated with biochemical pools in this study (Table 5).
We measured irradiance during our incubation time (4~5h) for phytoplankton productivity. This short
time of measured irradiance might be not enough to reflect and detect the change of biochemical
composition in phytoplankton with irradiance
The structure and composition of phytoplankton assemblages and species could have a
significant influence on the seasonal variation of biochemical composition. Although we did not
conduct a study of phytoplankton community structure, there is seasonal succession of phytoplankton
community structure in the bay. Previous studies showed that the dominant phytoplankton community
was diatoms and dominant diatom species were *Skeletonema spp.* during summer and winter in
Gwangyang bay (Choi et al., 1998; Baek et al., 2015). Kim et al. (2009) also reported that diatom and
dinoflagellate communities have experienced a considerable change because of increased nutrient
loadings from both domestic sewage and industrial pollution during summer. Therefore, the seasonal
change of phytoplankton species composition and community structure could lead to determining
different biochemical pools on seasonal basis.
However, other studies in different regions reported that environmental conditions, such as
temperature, nutrients and irradiance are more important controlling factors in biochemical
composition than variation of phytoplankton community and species composition (Lindqvist and
Lingnell, 1997; Suárez and Marañón, 2003). In this study, we also concluded that DIN from river-
input was a primary governing factor for the seasonal variation of biochemical composition of
phytoplankton in Gwangyang Bay as discussed above.
**4.4. Total FM and energy content of POM in a global context**
The annual average of FM was 434.5 µg L$^{-1}$ (S.D. = ± 175.5 µg L$^{-1}$) in this study. Since there
were no comparable data available in South Korea, we compared our results with other regions (Table





7), although they were conducted in different seasons and sampling depths. PRT contents in this study
were as high as in the Ross Sea (Fabiano and Puscceddue, 1998; Fabiano et al., 1999a), the Amundsen
Sea (Kim et al., 2015) and the Humboldt Current System (Isla et al., 2010). A similar range of LIP
contents was observed in Bedford Basin (Mayzaud et al., 1989), Yaldad Bay (Navarro et al., 1993)
and the Humboldt Current System (Isla et al., 2010). CHO contents were comparatively higher in this
study than other studies except Bedford Basin (Mayzaud et al., 1989) and Yaldad Bay (Navarro et al.,
1993). One of the highlights is that the calorific contents of FM were generally higher than those of
other areas except several regions. The FM values were comparatively higher than other regions such
as the northern Chuckchi Sea (Kim et al., 2014; Yun et al., 2014), Ross Sea (Fabiano et al., 1996;
Fabiano and Pusceddu, 1998; Fabiano et al., 1999a; Pusceddu et al., 1999), Amundsen Sea (Kim et al.,
2015) and the northern part of the East/Japan Sea (Kang et al., unpublished) or similar to the
Humboldt Current System which is known as an important spawning sites for pelagic fishes and the
highest abundance of anchovy eggs (Isla et al., 2010). Actually, the southern coastal sea (including our
study area) in Korea represents calm seas, an indented coastline, and numerous bays, which have
high diversities of habitat for fishes and shellfishes (Kwak et al., 2012) and give a favorable condition
for mariculture (Kwon et al., 2004). The high quantity of FM and the calorific contents of POM found
in this study reflected good nutritive conditions of primary food materials mainly provided by
phytoplankton for the maintenance of productive shellfish and fish populations in Gwangyang bay.
**5. Summary and Conclusion**
This study is the first report that was investigated the biochemical composition of POM on
seasonal basis in Gwangyang Bay, South Korea and we determined major controlling factors for
biochemical composition which is influenced by various environmental factors (Morris et al., 1974;
Smith and Morris, 1980; Rivkin and Voytek, 1987; Boëchat and Giani, 2008; Cuhel and Lean, 1987;
Mock and Kroon, 2002; Khotimchenko and Yakoleva, 2005; Ventura et al., 2008; Sterner et al. 1997).
Among different factors, temperature was positively correlated with LIP whereas negatively





correlated with PRT in this study (Table 5), which is consistent with previous works. In addition, we
found that PRT and LIP compositions were strongly correlated with DIN loading largely depending on
the amount of river-input from the Seomjin river which influences on PRT and LIP synthesis and
subsequently macromolecular composition of phytoplankton in Gwangyang bay. The concentrations
and the calorific contents of FM found in this study were relatively higher in comparison to previous
studies in various regions, which reflecting that POM (mainly from phytoplankton) provides a good
nutritive condition to maintain this highly productive estuarine ecosystem in Gwangyang bay.

Recently, significant environmental perturbations in their watersheds and externally from

climatic forcings have been reported in various estuaries (Wetz and Yoskowitz, 2013). More intense
but less frequent tropical cyclones are expected over the coming century (e.g., Elsner et al., 2008;
Knutson et al., 2010) and many changes in drought and flood cycles have been proceeding globally
(e.g., Min et al., 2011; Pall et al., 2011; Trenberth and Fasullo, 2012; Trenberth, 2012). The
cumulative effects of these perturbations could alter the quantity and quality of biochemical
composition of POM and cause subsequent changes in ecosystem structure and trophic dynamics in
estuaries (Cloern, 2001; Paerl et al., 2006; Rabalais et al., 2009; Wetz and Yoskowitz, 2013).
Therefore, continuous field measurements and observations on biochemical composition of POM   as
food quality are needed to monitor   for better understanding future response of marine ecosystem on
potential environmental perturbations in Gwangyang Bay.

**Acknowledgements**

This research was supported by "Long-term change of structure and function in marine

ecosystems of Korea" funded by the Ministry of Oceans and Fisheries, Korea.




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



**Table captions**
Table 1. Environmental factors and chl-*a* concentrations in Gwangyang bay during the research period

(- : no data).

Table 2. Rainfall and river input.
Table 3. $\delta^{13}$C values and C:N ratios of POM in Gwangyang bay (surface).
Table 4. Biochemical concentrations and composition, calorific values and contents in Gwangyang

bay (- : no data).

Table 5. Significant correlation coefficient (r) among proteins (PRT), lipids (LIP) and environmental

factors (ns ; no significance, **; *p*<0.01).

Table 6. Observed nutrient limitations during the study period.
Table 7. Comparison of biochemical quantity of POM, FM and the calorific contents.





**Figure captions**

Fig. 1. Sampling location in Gwangyang bay, Korea ; Maps of Korea (a), Southern Coastal Sea (b)

and main sampling stations (c).

Fig. 2. Seasonal variation of biochemical composition in Gwangyang bay.

Fig. 3. The positive relationship between river-input and protein composition.

Fig. 4. The inverse relationship between lipid compositions and protein compositions.




Table 1. Environmental factors and chl-*a* concentrations in Gwangyang Bay during the research period (- : no data).


| Year | Date | Irradiance ($\mu$mols m$^{-2}$ s$^{-1}$) | Station | Light depth (%) | Temperature (°C) | Salinity (‰) | Depth (m) | NH$_4$ ($\mu$M) | NO$_2$+NO$_3$ ($\mu$M) | SiO$_2$ ($\mu$M) | PO$_4$ ($\mu$M) | Chl-*a* ($\mu$g L$^{-1}$) |
|---|---|---|---|---|---|---|---|---|---|---|---|---|
| 2012 | April | 167.9 ± 133.5 | St.1 | 100 | 13.9 | 14.5 | 0 | 3.6 | 56.4 | 26.0 | 80.9 | 1.89 |
| | | (average ± S.D.) | | 30 | 13.3 | 25.6 | 1 | - | - | - | - | 1.95 |
| | | | | 1 | 13.5 | 28.0 | 3 | 2.4 | 16.0 | 9.8 | 0.2 | 2.08 |
| | | | St.4 | 100 | 15.0 | 24.4 | 0 | 2.6 | 15.1 | 16.3 | 0.2 | 1.81 |
| | | | | 30 | 13.6 | 31.4 | 1 | - | - | - | - | - |
| | | | | 1 | 12.3 | 32.9 | 5 | 1.9 | 2.1 | 2.1 | 0.1 | 2.03 |
| | | | St.5 | 100 | 12.6 | 31.7 | 0 | 3.1 | 9.5 | 7.1 | 0.3 | 2.07 |
| | | | | 30 | 12.3 | 31.6 | 1 | - | - | - | - | - |
| | | | | 1 | 12.2 | 32.4 | 5 | 3.0 | 6.4 | 5.1 | 0.3 | 2.04 |
| | June | 1158.1 ± 627.6 | St.2A | 100 | 22.9 | 27.6 | 0 | - | - | - | - | 1.77 |
| | | | | 30 | 22.8 | 27.6 | 1 | - | - | - | - | 0.76 |
| | | | | 1 | 22.9 | 28.7 | 3 | - | - | - | - | 0.76 |
| | | | St.4 | 100 | 23.6 | 31.5 | 0 | - | - | - | - | 1.00 |
| | | | | 30 | 22.6 | 31.9 | 3 | - | - | - | - | 1.67 |
| | | | | 1 | 22.1 | 32.3 | 11 | - | - | - | - | 1.02 |
| | August | 1320.0 ±316.9 | St.4 | 100 | 25.8 | 30.6 | 0 | 0.1 | 0.1 | 10.6 | 0.1 | 8.11 |
| | | | | 30 | 25.7 | 31.6 | 2 | - | - | - | - | 8.49 |
| | | | | 1 | 25.7 | 31.7 | 8 | 0.1 | 0.1 | 11.9 | 0.1 | 5.99 |
| | | | St.5 | 100 | 25.6 | 31.6 | 0 | 0.7 | 0.3 | 8.2 | 0.0 | 14.20 |
| | | | | 30 | 26.1 | 31.5 | 2 | - | - | - | - | 9.85 |
| | | | | 1 | 25.7 | 31.7 | 8 | 0.1 | 0.1 | 10.1 | 0.1 | 3.19 |
| | October | - | St.2A | 100 | 20.6 | 29.8 | 0 | 1.4 | 3.0 | 11.3 | 0.1 | 1.07 |
| | | | | 30 | 20.5 | 29.8 | 1 | - | - | - | - | 1.30 |
| | | | | 1 | 21.9 | 30.2 | 3 | 1.3 | 1.3 | 8.1 | 0.1 | 1.24 |
| | | | St.4 | 100 | 20.9 | 30.3 | 0 | 1.6 | 3.1 | 14.0 | 0.1 | 2.69 |
| | | | | 30 | 20.7 | 30.3 | 1 | - | - | - | - | 2.93 |
| | | | | 1 | 20.6 | 30.6 | 5 | 1.1 | 0.6 | 7.4 | 0.1 | 1.74 |
| | | | St.5 | 100 | 19.1 | 30.4 | 0 | 1.0 | 0.4 | 6.5 | 0.1 | 2.47 |
| | | | | 30 | 18.5 | 30.5 | 2 | - | - | - | - | 1.98 |
| | | | | 1 | 18.1 | 30.4 | 8 | 1.2 | 0.2 | 5.3 | 0.0 | 2.20 |






| Year | Date | Irradiance (µmols m$^{-2}$ s$^{-1}$) | Station | Light depth (%) | Temperature (°C) | Salinity (‰) | Depth (m) | NH$_4$ (µM) | NO$_2$+NO$_3$ (µM) | SiO$_2$ (µM) | PO$_4$ (µM) | Chl-$a$ (µg L$^{-1}$) |
|---|---|---|---|---|---|---|---|---|---|---|---|---|
| 2013 | January | 297.4 ± 310.5 | St.2A | 100 | 5.5 | 20.5 | 0 | 0.5 | 4.2 | 4.0 | 0.1 | 1.39 |
| | | | | 30 | 7.0 | 28.0 | 1 | - | - | - | - | 1.52 |
| | | | | 1 | 7.3 | 29.4 | 4 | 0.5 | 3.7 | 3.6 | 0.1 | 1.48 |
| | | | St.4 | 100 | 7.7 | 31.1 | 0 | 1.0 | 3.8 | 3.4 | 0.1 | 2.79 |
| | | | | 30 | 7.4 | 31.3 | 4 | - | - | - | - | 3.41 |
| | | | | 1 | 7.3 | 32.8 | 12 | 0.6 | 3.1 | 2.5 | 0.0 | 5.37 |
| | | | St.5 | 100 | 6.3 | 31.8 | 0 | 0.8 | 3.3 | 2.6 | 0.1 | 5.79 |
| | | | | 30 | 6.6 | 31.9 | 3 | - | - | - | - | 5.25 |
| | | | | 1 | 6.4 | 32.5 | 11 | 1.0 | 3.0 | 3.6 | 0.2 | 5.33 |
| | April | 1593.3 ± 414.5 | St.2A | 100 | 14.3 | 26.2 | 0 | 1.9 | 3.7 | 3.1 | 0.1 | 1.81 |
| | | | | 30 | 14.4 | 27.5 | 1 | - | - | - | - | 1.72 |
| | | | | 1 | 14.3 | 29.1 | 3 | 1.5 | 2.5 | 2.3 | 0.1 | 2.06 |
| | | | St.4 | 100 | 14.7 | 32.0 | 0 | 1.6 | 2.0 | 2.5 | 0.1 | 2.24 |
| | | | | 30 | 15.3 | 32.0 | 1 | - | - | - | - | 4.41 |
| | | | | 1 | 15.2 | 32.6 | 5 | 1.5 | 1.7 | 1.6 | 0.1 | 7.39 |
| | | | St.5 | 100 | 16.1 | 31.9 | 0 | 1.1 | 1.3 | 1.3 | 0.1 | 4.39 |
| | | | | 30 | 16.1 | 32.0 | 3 | - | - | - | - | 5.22 |
| | | | | 1 | 16.6 | 32.3 | 11 | 1.1 | 0.7 | 1.0 | 0.1 | 5.90 |





Table 2. Rainfall and river input

| Year | Date | Rainfall (mm) | River input ($10^6$ t) |
|------|------|---------------|------------------------|
| 2012 | April | 195.5 | 149.4 |
| | May | 44.4 | 148.9 |
| | June | 69.6 | 42.3 |
| | July | 235.8 | 223.3 |
| | August | 559.0 | 228.9 |
| | September | 360.1 | 447.2 |
| | October | 38.0 | 98.5 |
| | November | 52.5 | 83.4 |
| | December | 96.7 | 89.4 |
| 2013 | January | 15.6 | 79.3 |
| | February | 116.4 | 94.6 |
| | March | 79.9 | 91.5 |
| | April | 99.1 | 100.3 |





Table 3. $\delta^{13}$C values and C:N ratios of POM in Gwangyang Bay (surface)

| Year | Date | $\delta^{13}$C (‰) | C:N ($\mu$g $\mu$g$^{-1}$) |
|---|---|---|---|
| 2012 | April | -22.8 | 7.0 |
| | June | -23.1 | 6.8 |
| | August | -16.5 | 6.7 |
| | October | -17.1 | 6.9 |
| 2013 | January | -22.5 | 7.7 |
| | April | -23.1 | 6.8 |
| | (average ± S.D.) | -20.9 ± 3.2 | 7.0 ± 0.4 |





Table 4. Biochemical concentrations and composition, calorific values and contents in Gwangyang Bay (- : no data).

| Year | Date | Station | Light depth (%) | CHO ($\mu g\ L^{-1}$) | PRT ($\mu g\ L^{-1}$) | LIP ($\mu g\ L^{-1}$) | FM ($\mu g\ L^{-1}$) | CHO/FM (%) | PRT/FM (%) | LIP/FM (%) | Kcal $g^{-1}$ | Kcal $m^{-3}$ |
|---|---|---|---|---|---|---|---|---|---|---|---|---|
| 2012 | April | St.1 | 100 | 45.0 | 144.2 | 22.9 | 212.1 | 21.2 | 68.0 | 10.8 | 5.6 | 1.2 |
| | | | 30 | 53.1 | 218.6 | 51.9 | 323.6 | 16.4 | 67.6 | 16.0 | 5.9 | 1.9 |
| | | | 1 | 53.1 | 220.4 | 84.2 | 357.6 | 14.8 | 61.6 | 23.5 | 6.2 | 2.2 |
| | | St.4 | 100 | 14.2 | 128.1 | 28.6 | 170.9 | 8.3 | 74.9 | 16.7 | 6.1 | 1.0 |
| | | | 30 | 50.0 | 155.1 | 21.4 | 226.5 | 22.1 | 68.5 | 9.4 | 5.6 | 1.3 |
| | | | 1 | 20.2 | 146.0 | 37.3 | 203.5 | 9.9 | 71.8 | 18.3 | 6.1 | 1.2 |
| | | St.5 | 100 | 60.2 | 198.0 | 143.0 | 401.2 | 15.0 | 49.3 | 35.7 | 6.7 | 2.7 |
| | | | 30 | 132.4 | 198.0 | 42.8 | 373.2 | 35.5 | 53.1 | 11.5 | 5.5 | 2.0 |
| | | | 1 | 146.7 | 265.3 | 210.0 | 622.1 | 23.6 | 42.7 | 33.8 | 6.5 | 4.1 |
| | June | St.2A | 100 | 170.7 | 99.7 | 233.5 | 503.8 | 33.9 | 19.8 | 46.3 | 6.9 | 3.5 |
| | | | 30 | 135.5 | 108.0 | 251.9 | 495.4 | 27.3 | 21.8 | 50.9 | 7.2 | 3.5 |
| | | | 1 | 163.5 | 85.0 | 225.1 | 473.7 | 34.5 | 17.9 | 47.5 | 6.9 | 3.3 |
| | | St.4 | 100 | 99.1 | 44.6 | 199.5 | 343.2 | 28.9 | 13.0 | 58.1 | 7.4 | 2.5 |
| | | | 30 | 133.4 | 142.4 | 203.5 | 479.3 | 27.8 | 29.7 | 42.4 | 6.8 | 3.3 |
| | | | 1 | 91.6 | 110.8 | 232.3 | 434.6 | 21.1 | 25.5 | 53.5 | 7.3 | 3.2 |
| | August | St.4 | 100 | 69.3 | 73.9 | 213.5 | 356.7 | 19.4 | 20.7 | 59.9 | 7.6 | 2.7 |
| | | | 30 | 61.2 | 56.5 | 173.8 | 291.5 | 21.0 | 19.4 | 59.6 | 7.6 | 2.2 |
| | | | 1 | 127.2 | 77.9 | 162.2 | 367.3 | 34.6 | 21.2 | 44.2 | 6.8 | 2.5 |
| | | St.5 | 100 | 155.5 | 289.4 | 204.7 | 649.6 | 23.9 | 44.6 | 31.5 | 6.4 | 4.2 |
| | | | 30 | 412.3 | 102.0 | 401.4 | 915.7 | 45.0 | 11.1 | 43.8 | 6.6 | 6.1 |
| | | | 1 | 83.3 | 22.8 | 228.3 | 334.4 | 24.9 | 6.8 | 68.3 | 7.9 | 2.6 |
| | October | St.2A | 100 | 71.0 | 82.2 | 104.1 | 257.3 | 27.6 | 32.0 | 40.5 | 6.7 | 1.7 |
| | | | 30 | 42.7 | 62.4 | 100.3 | 205.4 | 20.8 | 30.4 | 48.8 | 7.2 | 1.5 |
| | | | 1 | 74.3 | 111.6 | 98.5 | 284.4 | 26.1 | 39.2 | 34.6 | 6.5 | 1.9 |
| | | St.4 | 100 | 51.6 | 105.2 | 105.3 | 262.2 | 19.7 | 40.1 | 40.2 | 6.8 | 1.8 |
| | | | 30 | 119.4 | 121.9 | 144.4 | 385.6 | 31.0 | 31.6 | 37.4 | 6.6 | 2.5 |
| | | | 1 | 78.5 | 169.0 | 134.4 | 381.9 | 20.6 | 44.2 | 35.2 | 6.6 | 2.5 |
| | | St.5 | 100 | 37.2 | 70.0 | 86.5 | 193.6 | 19.2 | 36.1 | 44.7 | 7.0 | 1.4 |
| | | | 30 | 42.3 | 92.5 | 112.0 | 246.7 | 17.2 | 37.5 | 45.4 | 7.1 | 1.7 |
| | | | 1 | 33.9 | 108.4 | 97.3 | 239.7 | 14.2 | 45.2 | 40.6 | 6.9 | 1.7 |




| Year | Date | Station | Light depth (%) | CHO ($\mu g\ L^{-1}$) | PRT ($\mu g\ L^{-1}$) | LIP ($\mu g\ L^{-1}$) | FM ($\mu g\ L^{-1}$) | CHO/FM (%) | PRT/FM (%) | LIP/FM (%) | Kcal $g^{-1}$ | Kcal $m^{-3}$ |
|---|---|---|---|---|---|---|---|---|---|---|---|---|
| 2013 | January | St.2A | 100 | 150.3 | 139.3 | 115.5 | 405.2 | 37.1 | 34.4 | 28.5 | 6.1 | 2.5 |
| | | | 30 | 347.0 | 131.1 | 109.2 | 587.3 | 59.1 | 22.3 | 18.6 | 5.4 | 3.2 |
| | | | 1 | 331.3 | 127.1 | - | - | - | - | - | - | - |
| | | St.4 | 100 | 171.6 | 164.0 | - | - | - | - | - | - | - |
| | | | 30 | 183.5 | 168.7 | 139.7 | 491.9 | 37.3 | 34.3 | 28.4 | 6.1 | 3.0 |
| | | | 1 | 115.9 | 182.3 | 107.1 | 405.2 | 28.6 | 45.0 | 26.4 | 6.2 | 2.5 |
| | | St.5 | 100 | 113.6 | 212.0 | 133.4 | 459.0 | 24.7 | 46.2 | 29.1 | 6.3 | 2.9 |
| | | | 30 | 264.1 | 204.8 | 120.5 | 589.4 | 44.8 | 34.8 | 20.4 | 5.7 | 3.4 |
| | | | 1 | 99.3 | 195.5 | 104.2 | 399.0 | 24.9 | 49.0 | 26.1 | 6.2 | 2.5 |
| | Apirl | St.2A | 100 | 237.7 | 262.9 | 189.9 | 690.5 | 34.4 | 38.1 | 27.5 | 6.1 | 4.2 |
| | | | 30 | 185.5 | 308.0 | 198.7 | 692.3 | 26.8 | 44.5 | 28.7 | 6.3 | 4.3 |
| | | | 1 | 274.8 | 382.4 | 180.3 | 837.5 | 32.8 | 45.7 | 21.5 | 5.9 | 4.9 |
| | | St.4 | 100 | 115.0 | 141.9 | 181.4 | 438.4 | 26.2 | 32.4 | 41.4 | 6.8 | 3.0 |
| | | | 30 | 116.4 | 187.0 | 191.0 | 494.5 | 23.5 | 37.8 | 38.6 | 6.7 | 3.3 |
| | | | 1 | 205.2 | 222.1 | 185.7 | 612.9 | 33.5 | 36.2 | 30.3 | 6.2 | 3.8 |
| | | St.5 | 100 | 160.4 | 176.3 | 289.1 | 625.7 | 25.6 | 28.2 | 46.2 | 7.0 | 4.4 |
| | | | 30 | 146.9 | 217.8 | 253.3 | 618.0 | 23.8 | 35.2 | 41.0 | 6.8 | 4.2 |
| | | | 1 | 171.3 | 204.9 | 272.6 | 648.8 | 26.4 | 31.6 | 42.0 | 6.8 | 4.4 |




Table 5. Significant correlation coefficient (r) among proteins (PRT), lipids (LIP) and environmental factors (ns ; no significance, **; $p<0.01$).

| Variables | r | p | n |
|---|---|---|---|
| %PRT $\times$ Temp. | - 0.52 | ** | 46 |
| %LIP $\times$ Temp. | 0.72 | ** | 46 |
| %PRT $\times$ $NH_4$ | 0.69 | ** | 28 |
| %LIP $\times$ $NH_4$ | -0.59 | ** | 28 |
| %PRT $\times$ $NO_2+NO_3$ | 0.54 | ** | 28 |
| %LIP $\times$ $NO_2+NO_3$ | -0.53 | ** | 28 |
| %PRT $\times$ River-input | 0.84 | ** | 46 |
| %LIP $\times$ River-input | -0.63 | ** | 46 |
| $NH_4$ $\times$ River-input | 0.91 | ** | 28 |
| $NO_2+NO_3$ $\times$ River-input | 0.55 | ** | 28 |
| %PRT $\times$ %LIP | -0.81 | ** | 46 |
| %PRT $\times$ Irradiance | -0.22 | ns | 39 |
| %LIP $\times$ Irradiance | 0.24 | ns | 39 |





Table 6. Observed nutrient limitations during the study period.

| Year | Date | Based on absolute concentrations (μM) | | | | Based on molar ratios | | |
|------|------|------|------|------|------------|------|------|------------|
| | | DIN | SiO$_2$ | PO$_4$ | Limitation | Si:N | Si:P | Limitation |
| 2012 | April | 20.3 ± 20.2 | 11.1 ± 8.8 | 13.6 ± 32.9 | nd | 0.6 ± 0.2 | 37.5 ± 36.9 | nd |
| | June | - | - | - | - | - | - | - |
| | August | 0.4 ± 0.4 | 10.2 ± 1.5 | 0.1 ± 0.0 | N, P | 42.7 ± 23.7 | 173.4 ± 56.5 | N, P |
| | October | 2.7 ± 1.5 | 8.8 ± 3.3 | 0.1 ± 0.0 | P | 3.6 ± 0.8 | 142.2 ± 74.0 | N |
| 2013 | January | 4.2 ± 0.4 | 3.3 ± 0.6 | 0.1 ± 0.1 | P | 0.8 ± 0.1 | 50.6 ± 41.4 | nd |
| | April | 3.4 ± 1.3 | 2.0 ± 0.8 | 0.1 ± 0.0 | P | 0.6 ± 0.1 | 15.5 ± 5.5 | Si |





Table 6. Comparison of biochemical quantity of POM, FM and the calorific contents.

| Regions (depth) | | PRT (µg l⁻¹) | LIP (µg l⁻¹) | CHO (µg l⁻¹) | FM (µg l⁻¹) | Kcal m⁻³ (average ± S.D.) | Authors |
|---|---|---|---|---|---|---|---|
| | Gwangyang Bay, South Korea (Euphotic depth) | 23-382 | 21-401 | 14-412 | 171-916 | 2.8 ± 1.1 | This study |
| Arctic regions | Bedford Basin, Canada(2.5 m) | 200-650 | 130-440 | 160-630 | 660-1570 | | Mayzaud et al. (1989) |
| | Logy Bay, Newfoundland (6 m) | 80-740 | 20-75 | 8-120 | 130-1030 | 2.7 ± 2.8 | Navarro & Thompson (1995) |
| | The Northern Chukchi Sea, 2011 (Euphotic depth) | 1-86 | 50-105 | 22-147 | 94-246 | 1.0 ± 0.2 | Kim et al. (2015) |
| | The Northern Chukchi Sea, 2012 (Euphotic depth) | 9-183 | 37-147 | 16-253 | 90-373 | 1.2 ± 0.2 | Yun et al. (2015) |
| Antarctic regions | Pacific Sector Antarctic Ocean (0-1500 m) | 14-100 | 3-60 | 3-66 | 25-220 | | Tanoue (1985) |
| | Off Princess Astrid Coast, Antarctica (0-100m) | 24-200 | 15-174 | 22-147 | 148-393 | | Dhargalkar et al. (1996) |
| | Ross Sea, Antarctica (10m) | 11-402 | 91 | 91-187 | 193-680 | 2.6 ± 1.8 | Fabiano and Pusceddu (1998) |
| | Ross Sea, Antarctica (0-200 m) | 40-406 | 18-115 | 22-251 | 110-660 | | Fabiano et al. (1999) |
| | Terra Nova Bay, Antarctica (0-750 m) | 10-620 | 2-77 | 8-144 | 19-885 | 1.3 ± 1.0 | Fabiano et al. (1996) |
| | Terra Nova Bay, Antarctica (under pack ice) | 96-201 | 38-112 | 10-68 | 145-382 | 1.7 ± 1.1 | Pusceddue et al. (1999) |
| | Amundsen Sea (Euphotic depth) | 6-396 | 13-37 | 3-216 | 43-639 | 1.2 ± 0.8 | Kim et al. (2015) |
| Other regions | W-Mediterranean (0-200 m) | 72-105 | 37-51 | 33-88 | 143-246 | | Fabiano et al. (1984) |
| | W-Mediterranean submarine cave (10m) | 4-77 | 4-104 | 1-75 | 15-220 | 0.4 ± 0.2 | Fichez (1991b) |
| | Mediterranean seagrass (4 m) | 25-135 | 50-180 | 40-110 | 125-395 | | Danovaro et al. (1998) |
| | Ligurian Sea (10 m) NW-Mediterranean | 32-107 | 21-140 | 21-131 | 74-378 | 1.5 ± 1.4 | Danovaro & Fabiano (1997) |
| | Mediterranean (30m) | 70-90 | 90-110 | 10-20 | 177-213 | 1.4 ± 0.2 | Modica et al. (2006) |
| | Cretan Sea (0-1500 m) | 7-92 | 4-63 | 13-149 | 54-200 | 0.6 ± 0.2 | Danovaro et al. (2000) |
| | Bay of Biscay, 2000 (0-30m) | 109-2426 | 26-2037 | 2-345 | 961 (a.v.) | 6.7 ± 5.0 | Díaz et al. (2007) |
| | Yaldad Bay, Chile (10 cm a.b.) | 300-2250 | 30-560 | 50-1050 | 3310-2960 | 10.0 ± 10.9 | Navarro et al. (1993) |
| | The Humboldt current system, Northern Chile (5-89m) | 40-470 | 60-390 | 70-510 | 24-1282 | 3.5 ± 3.3 | Isla et al. (2010) |
| | Magellan Strait (0-50m) | 60-150 | 30-70 | 20-40 | 110-256 | 1.0 ± 0.5 | Fabiano et al. (1999) |
| | The northern part of the East Sea (Euphotic depth) | 28-425 | 12-180 | 19-206 | 109-810 | 1.5 ± 0.6 | Kang et al. (unpublished) |






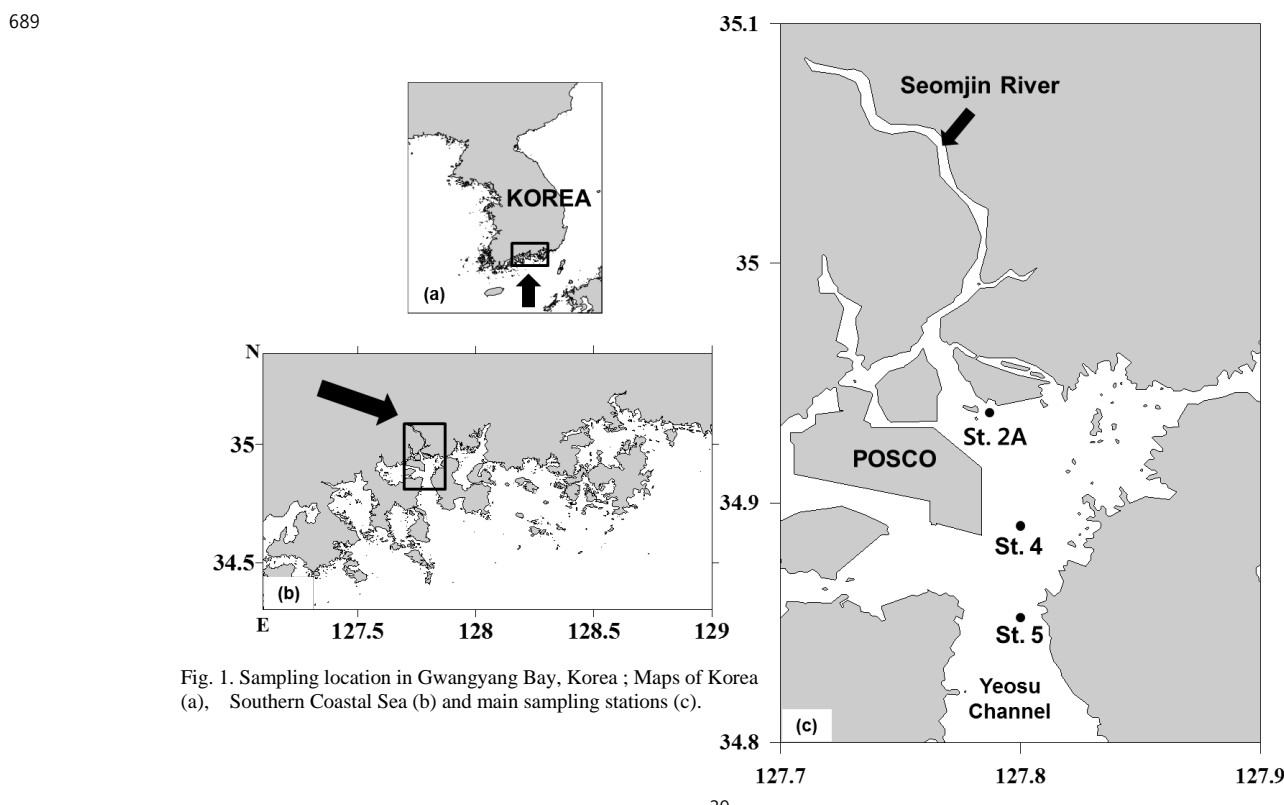

Fig. 1. Sampling location in Gwangyang Bay, Korea ; Maps of Korea (a), Southern Coastal Sea (b) and main sampling stations (c).




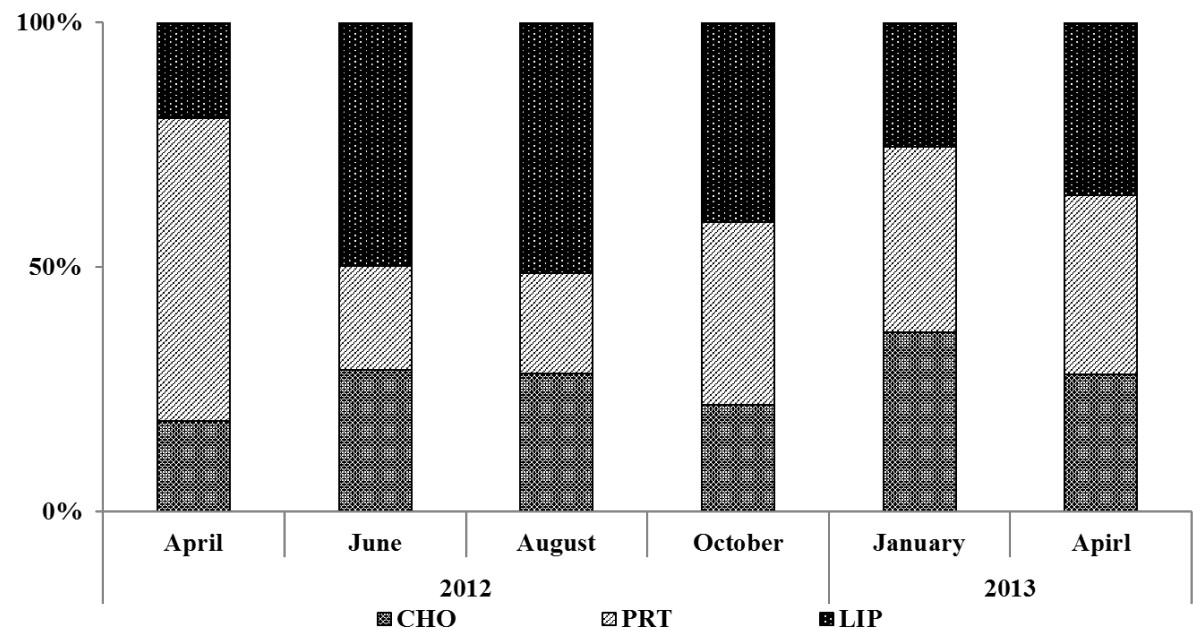

Fig. 2. Seasonal variation of biochemical composition in Gwangyang Bay.





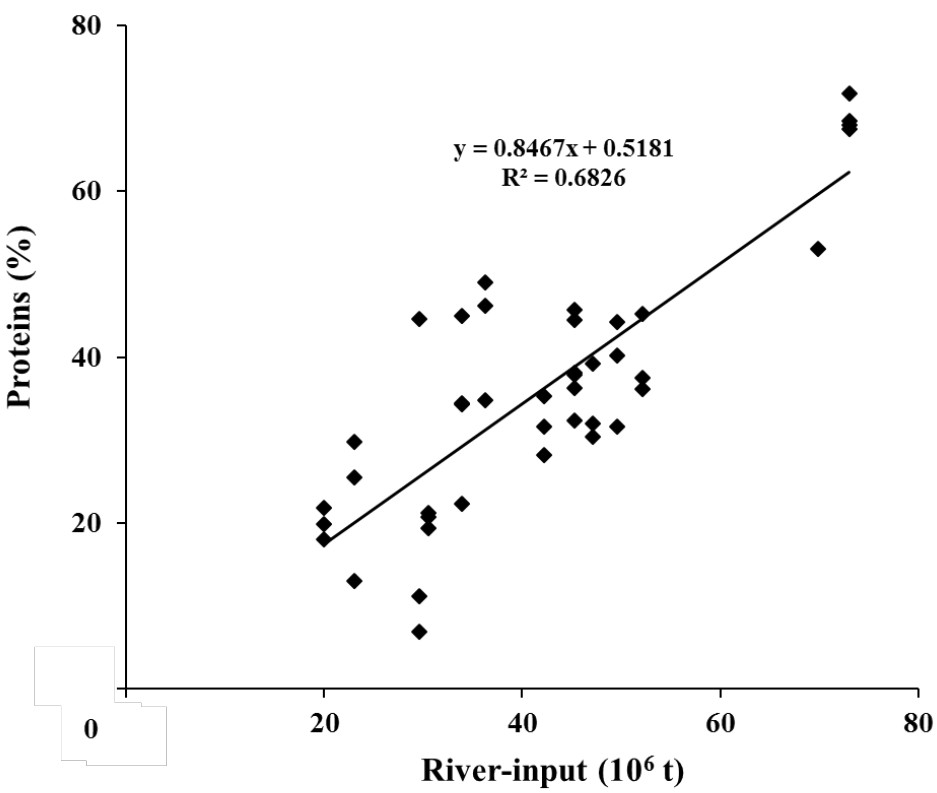

Fig. 3. The positive relationship between river-input and protein composition.






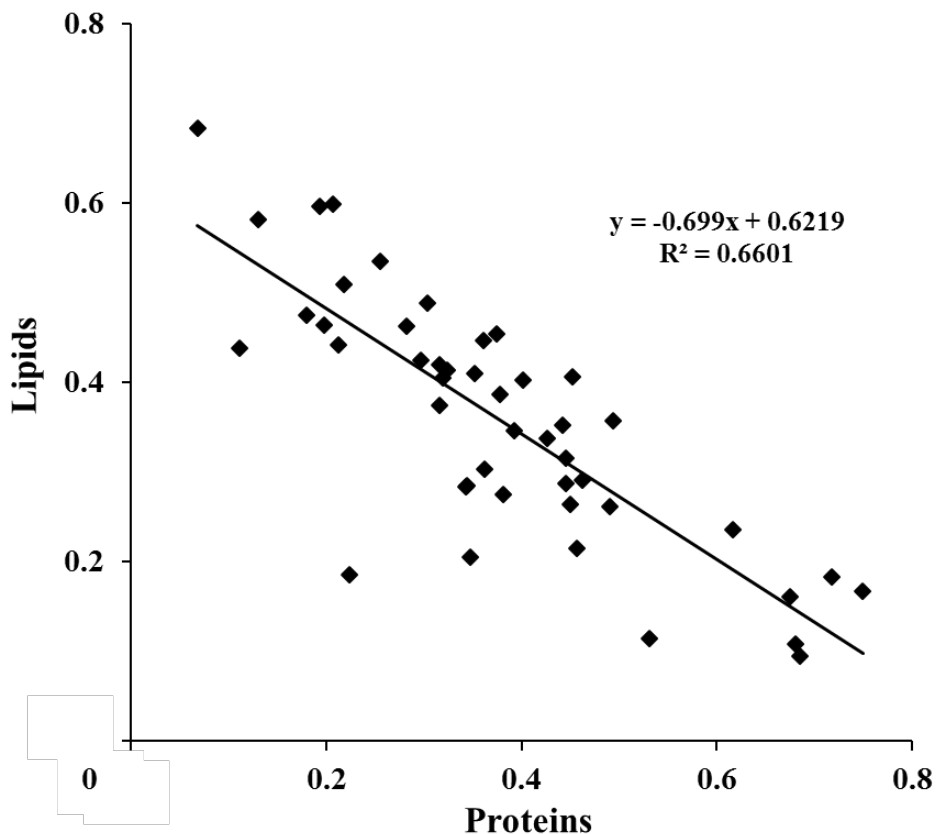

Fig. 4. The inverse relationship between lipid compositions and protein compositions