# Peer review of "The effects of different environmental factors on biochemical composition of particulate organic matters in Gwangyang Bay, South Korea"

_Biogeosciences, 2016_

## Referee Comment (RC1) · Anonymous Referee #1 · 16 Sep 2016

Review of Biogeosciences Discuss., doi:10.5194/bg-2016-347 The effects of different environmental factors on biochemical composition of particulate organic matter in Gwangyang Bay, South Korea written by Jang Han Lee, Dabin Lee, Jae Joong Kang, Hui Tae Joo, Jae Hyung Lee, Ho Won Lee, So Hyun Ahn and Sang Heon Lee In the submitted article the authors analyzed seasonal changes of the biochemical composition (proteins, lipids, carbohydrates) of the particulate organic matter and linked it to environmental factors in order to determine the major environmental factor influencing the changes of biochemical composition and the origin of particulate organic carbon. In general, the paper has a scientific potential and some parts of the paper are fairly

discussed (biochemical composition) and linked to the relevant literature. However, some parts of the sections Materials and methods, Results and Discussion are not clearly outlined or missed important information that complicate understanding of the text and question the purpose of applied experimental design. The conclusions are mostly repeating of the results so it should be also rewritten and the last paragraph omitted, it is too general and does not contain the conclusion of the paper. The major revision and resubmission is recommended. The experimental design was based on three different light intensity depths along three stations in bay and all results were pooled together on the monthly basis since no significant differences between vertical and spatial distributions were found. It was mentioned in the Material and methods that some statistical tests (ANOVA, t-test) were used, but it is not clear which test they used, where and which parameters they tested and how (there is 1 concentration per 1 depth at 1 station). The authors used very often describing results the word significant but did not specified the name of test, F-value or t-value. Details and reference about determination and/or calculation of the 30% and 1% of the photon flux based on Secchi disc depths should be added. It was only mentioned that the samples were incubated and later on in discussion it was written that the incubation time was too short? Further on, the light intensity and its impact (or no impact) on the biochemical composition is not discussed, particularly considering 10 times difference in light intensity between April 2012 and April 2013. These findings should be discussed with regard to a body of literature in which the influence of light was investigated and found. In the Table 1 there is irradiance expressed as ave±S.D.; I wonder if given average contains measurements from all stations on the day of sampling? Details about particulate organic carbon and nitrogen analysis such as volume of filtered water and station where the sample was taken should be added (only one result per month was presented). This is very important since the origin of POM is not typical for the estuaries. It is very interesting that riverine terrestrially derived organic matter is not an important component of the particulate organic matter in the Gwangyang Bay system, which has a large river runoff. One would expect partly organic matter of a terrestrial origin and not such clear

phytoplankton fingerprint since the water column is very turbid and euphotic layer very thin (3- 11 m). Also this peculiarity and these results should be discussed and compared with other estuaries like the authors did for biochemical composition. Nutrient limitation, the use of the ratios (lines 301-305): it is not clear why the authors use for the interpretation of phosphorus and nitrogen limitation only the ratios with dissolved silica (DSi) and not between these two components (N: P). If it was not a random error, the reference should be added for listed criteria. Anyway, in criteria b) for nitrogen limitation instead of DSi:DIP ratio >16 should stand < 16, if it was presumed that DSi and DIN appear in similar concentrations, though not always the case. References: Listed but not cited in the text: Adolf and Harding, 2006; Choi and Noh, 1998; De Oliveira et al 1999; Julian and David, 1966; Kim et al., 2016 Cited in the text but not listed: Choi et al., 1998; Kim et al., 2014; Kwon et al, 2001; Marsh and Weinstein 1966; Paerl et al., 2006; Yun et al., 2014 Cited or listed with different year of publication: Pirt 1975 (cited in the text), listed in references as Pirt 1976 Some references are written in uppercase. To the references published in the same year a, b should be added In Tables 1 and 4 in April 2012 appears st. 1 which is not marked on the map (Fig.1)

---

## Short Comment (SC1) · 27 Sep 2016

General Comments The manuscript presents the seasonal variation of biochemical composition of POM in Bay. The author shows the major controlling factor for them based on statistical analysis. Overall, I found the paper to be sound and believe that it contains valuable data in understanding the characteristics of POM and their contribution to coastal ecosystem as basic food source. I think that the paper is worthy of publication for BGS after minor revisions are made, while there are a few areas that need improvement.

Major comment and corrections

1. Page 12, Line 258-278: The author showed $\delta$13C value and carbon to nitrogen ratio in surface, in order to find the origin of POM. I think that the contribution of benthic microalgae to POM could be large and significant, since the study area is located in coastal area and extremely turbid condition related to freshwater input or tidal cycles or wind. Therefore, many amounts of benthic microalgae could be included to POM through the resuspension, especially during high river input. Indeed, Table 3 shows the lower 13C value in August.

2. Pages 13-14, Line 301-304: For the criteria of their moral ratios among dissolved inorganic nutrients, I wonder could it be applied in coastal area. I think that the status of nutrient limitation in phytoplankton could be different between open oceans and coastal area.

3. Page 15, Line 335-344: As the author discussed, I think that the composition of phytoplankton assemblages and species could be closely related to seasonal variation of biochemical composition. High nitrogen supply during river-input increased season could lead to different phytoplankton composition. For example, the large sized phytoplankton (such as diatom) could be thrived in that condition, since the large phytoplankton could grow best and dominate under eutrophic condition. According to Fernandez et al. (1994), the carbon allocation into different biochemical pools were different depending on dominant phytoplankton group. For example, the carbon allocation into lipids was higher under the dominance of flagellates, whereas the lower lipid synthesis was observed in the dominance of diatoms. Therefore, the seasonally different phytoplankton composition related to nutrient input could affect to the different biochemical composition in the region.

4. In figure 3, the author shows positive relationship between river input and protein composition. However, I didn't find the positive relationship between them, based on comparison with table 2 and figure 2. For example, the protein composition in August

was lowest, although the rive input was considerably high. In addition, the protein composition from October in 2012 to April in 2013 was higher than that in August, even though the lower river inputs were recorded.

Minor corrections

1. Pages 8-9, Line 175-186: The position of some sentences needs to be corrected. For example, the results about irradiance and chl-a are shown in Table 1 (it is explained in line 178-186). The results for rainfall and river-input are indicated in former position (in line 175-178), although they are shown in Table 2.

2. Page 9, Line 195-197: The author found that there were no significant differences in spatial distribution of POM. However, the protein composition in station 2A (is closest to the River) might be higher than in station 4 and 5, since there is the large effect of river-input on the biochemical composition in this study.

―――――――――――――――――――――

---

## Author Comment (AC2) · 20 Oct 2016

Dr. Yun (misunyun@pusan.ac.kr)

General Comments The manuscript presents the seasonal variation of biochemical composition of POM in Bay. The author shows the major controlling factor for them based on statistical analysis. Overall, I found the paper to be sound and believe that it contains valuable data in understanding the characteristics of POM and their contribution to coastal ecosystem as basic food source. I think that the paper is worthy of publication for BGS after minor revisions are made, while there are a few areas that need improvement.

Major comment and corrections 1. Page 12, Line 258-278: The author showed _13C value and carbon to nitrogen ratio in surface, in order to find the origin of POM. I think that the contribution of benthic microalgae to POM could be large and significant, since the study area is located in coastal area and extremely turbid condition related to freshwater input or tidal cycles or wind. Therefore, many amounts of benthic microalgae could be included to POM through the resuspension, especially during high river input. Indeed, Table 3 shows the lower 13C value in August. =>We discussed on potential contributions of benthic microalgae on POM in line 300-302, page 13.

2. Pages 13-14, Line 301-304: For the criteria of their moral ratios among dissolved inorganic nutrients, I wonder could it be applied in coastal area. I think that the status of nutrient limitation in phytoplankton could be different between open oceans and coastal area. =>Actually, the criteria of the moral ratios can be applied in coastal area based on several papers as we referred in our discussion (e.g., Roelke et al., 1999).

3. Page 15, Line 335-344: As the author discussed, I think that the composition of phytoplankton assemblages and species could be closely related to seasonal variation of biochemical composition. High nitrogen supply during river-input increased season could lead to different phytoplankton composition. For example, the large sized phytoplankton (such as diatom) could be thrived in that condition, since the large phytoplankton could grow best and dominate under eutrophic condition. According to Fernandez et al. (1994), the carbon allocation into different biochemical pools were different depending on dominant phytoplankton group. For example, the carbon allocation into lipids was higher under the dominance of flagellates, whereas the lower lipid synthesis was observed in the dominance of diatoms. Therefore, the seasonally different phytoplankton composition related to nutrient input could affect to the different biochemical composition in the region. =>Yes, the seasonal compositions of phytoplankton could lead different biochemical compositions. We discussed on that issues in line 380-389, page 15-16.

4. In figure 3, the author shows positive relationship between river input and protein composition. However, I didn't find the positive relationship between them, based on comparison with table 2 and figure 2. For example, the protein composition in August was lowest, although the rive input was considerably high. In addition, the protein composition from October in 2012 to April in 2013 was higher than that in August, even though the lower river inputs were recorded. =>Actually, the river input data in Table 2 are monthly integrated river inputs to show monthly patterns of river input and rainfall. In Figure 3, river-inputs were integrated from 20 days prior to our sampling dates since phytoplankton productivity is recovered after 20 days after rainfall in Gwangyang Bay according to Min et al. (2011). We added this explanation in line 100-102, page 5 to make it clear.

Minor corrections 1. Pages 8-9, Line 175-186: The position of some sentences needs to be corrected. For example, the results about irradiance and chl-a are shown in Table 1 (it is explained in line 178-186). The results for rainfall and river-input are indicated in former position (in line 175-178), although they are shown in Table 2. =>We rearranged the sentences in line 190-194, page 9. 2. Page 9, Line 195-197: The author found that there were no significant differences in spatial distribution of POM. However, the protein composition in station 2A (is closest to the River) might be higher than in station 4 and 5, since there is the large effect of river-input on the biochemical composition in this study. =>We did ANOVA test for each depth from 3 stations based on an assumption of no spatial difference and another ANOVA test for a spatial difference by pooling of 3 light depths at one station and comparing each station based on an assumption of no difference in light depths. But, we found to realize that there are statistical errors by doing that. So, we deleted no significant differences between vertical and spatial distributions in our text. The station 2A (is closest to the River) might be the largest effect of river-input but different effects of river-input could be different depending on water circulation, tidal currents, winds, and etc as well as distance from the river in Gwangyang Bay. At this point, we can not determine how much effect at each station from river inputs but the station 2A could have more proteins than others if they have more influence from river inputs based on Table 1 and Fig. 3.

Please also note the supplement to this comment:
http://www.biogeosciences-discuss.net/bg-2016-347/bg-2016-347-AC2-supplement.pdf

**Supplement:**

[revised manuscript text omitted]

---

## Referee Comment (RC2) · Anonymous Referee #2 · 30 Jan 2017

The manuscript, "The effects of different environmental factors on biochemical composition of particulate organic matter in Gwangyang Bay, South Korea" written by Lee et al., describes the seasonal changes of the biochemical composition such as proteins, lipids, and carbohydrates of the particulate organic matter and investigates the major environmental controlling factors for the changes of biochemical composition. This paper is very interesting to present seasonal biochemical composition of particulate organic matter responding to various environmental factors and the food quantity assessed in Gwangyang Bay since previous studies have focused mainly on different biochemical compositions as the authors mentioned that. The present study has

scientific merits and originality in that: 1. the topic, "biochemical composition of particulate organic matter as indicators of food quality and quantity", is very intriguing enough to draw much attention for understanding marine ecosystem especially here in Gwanyang Bay; 2. the study is one of few studies that employed seasonal biochemical composition of particulate organic matter responding to various environmental factors; 3. authors found that river-derived dissolved inorganic nitrogen loading is a major governing factor for determining the biochemical composition in a natural bay system which implies that man-made artificial dams could cause a serious disturbance in the ecosystem.

Overall, I would recommend publication of this manuscript for Biogeosciences after some minor revisions. I hope to see authors undertake revisions in an appropriate manner because I really want to see the final version of this paper in print.

Some minor comments are listed below: -. Be consistent with Gwangyang Bay or bay throughout the text. -. Be consistent with average or mean in the results. -. (Section 2.1) What time did usually authors make samplings? I'm wondering if there is tidal influence and/or diurnal changes? -. (line 76) indicate t (ton ?). -. (lines 83-83) Describe hot to determine 3 light intensities from secchi disk. -. (line 157 in section 2.5) Describe what Winberg (1971) is. -. (3. Results) Indicate statistics results in the result section. -. (Section 4.1) The first sentence in 4.1 section is repeated as the result section of 3.1. Remove it. -. (Table 6) No description for Table 6 is shown in the result section. Describe it in the result section before the discussion on that in the discussion section. -. (Section 4.4) The first sentence in 4.4 belongs to the result. -. (Table 7) Be consistent with the unit in Table 7 and the text. -. The conclusion section is needed to be revised since some parts in the conclusion repeat some results.

---

## Author Response (AR1)

Review of Biogeosciences Discuss., doi:10.5194/bg-2016-347 The effects of different environmental factors on biochemical composition of particulate organic matter in Gwangyang Bay, South Korea written by Jang Han Lee, Dabin Lee, Jae Joong Kang, Hui Tae Joo, Jae Hyung Lee, Ho Won Lee, So Hyun Ahn and Sang Heon Lee In the submitted article the authors analyzed seasonal changes of the biochemical composition (proteins, lipids, carbohydrates) of the particulate organic matter and linked it to environmental factors in order to determine the major environmental factor influencing the changes of biochemical composition and the origin of particulate organic carbon.

In general, the paper has a scientific potential and some parts of the paper are fairly discussed (biochemical composition) and linked to the relevant literature. However, some parts of the sections Materials and methods, Results and Discussion are not clearly outlined or missed important information that complicate understanding of the text and question the purpose of applied experimental design. The conclusions are mostly repeating of the results so it should be also rewritten and the last paragraph omitted, it is too general and does not contain the conclusion of the paper. The major revision and resubmission is recommended.

➔We revised each section throughout the manuscript, deleted most of repeating results in conclusions and revised carefully our manuscript based on referee # 1 comments as below.

The experimental design was based on three different light intensity depths along three stations in bay and all results were pooled together on the monthly basis since no significant differences between vertical and spatial distributions were found. It was mentioned in the Material and methods that some statistical tests (ANOVA, t-test) were used, but it is not clear which test they used, where and which parameters they tested and how (there is 1 concentration per 1 depth at 1 station).

➔ We did ANOVA test for each depth from 3 stations based on an assumption of no spatial difference and another ANOVA test for a spatial difference by pooling of 3 light depths at one station and comparing each station based on an assumption of no difference in light depths. But, we found to realize that there are statistical errors by doing that. So, we deleted no significant differences between vertical and spatial distributions.

The authors used very often describing results the word significant but did not specified the name of test, F-value or t-value.

➔We revised our results.

Details and reference about determination and/or calculation of the 30% and 1% of the photon flux based on Secchi disc depths should be added.

➔We added details and reference for light depth determination in line 87-90, page 4-5.

It was only mentioned that the samples were incubated and later on in discussion it was written that the incubation time was too short?

➔Our main purpose of the PAR measurements was calculating hourly primary productivity executed for 4~5 hours as a parallel study. Therefore, the irradiance values measured in this study were not representative for our sampling periods. We mentioned this in the method section in line 92-97, page 5 and further discussed on the issue in line 370-379, page 16.

Further on, the light intensity and its impact (or no impact) on the biochemical composition is not discussed, particularly considering 10 times difference in light intensity between April 2012 and April 2013. These findings should be discussed with regard to a body of literature in which the influence of light was investigated and found.

➜We added the discussion on the light intensity impact on the biochemical compositions, especially 10 times difference in light intensity between April 2012 and April 2013 in line 370-379, page 16.

In the Table 1 there is irradiance expressed as ave±S.D.; I wonder if given average contains measurements from all stations on the day of sampling?

➜We measured irradiance one time per each cruise at every 30 seconds during the incubation hours for primary productivity executed for 4~5 hours during day time around local noon time. So, ave±S.D. values in Table 1 are averages from every 30 seconds for 4-5 hours a day each season. We described the details in line 95-97, page 5.

Details about particulate organic carbon and nitrogen analysis such as volume of filtered water and station where the sample was taken should be added (only one result per month was presented). This is very important since the origin of POM is not typical for the estuaries.

➜We measured POC, PON, and $\delta^{13}$C of POM collected from surface at the 3 stations at every sampling time. They varied but did not show large differences in POC, PON, and $\delta^{13}$C among the different stations. We described the sampling details in line 114-120, page 6.

It is very interesting that riverine terrestrially derived organic matter is not an important component of the particulate organic matter in the Gwangyang Bay system, which has a large river runoff. One would expect partly organic matter of a terrestrial origin and not such clear phytoplankton fingerprint since the water column is very turbid and euphotic layer very thin (3- 11 m). Also this peculiarity and these results should be discussed and compared with other estuaries like the authors did for biochemical composition.

➜We further discussed on the issue in line 300-310, page 13-14.

Nutrient limitation, the use of the ratios (lines 301-305): it is not clear why the authors use for the interpretation of phosphorus and nitrogen limitation only the ratios with dissolved silica (DSi) and not between these two components (N: P). If it was not a random error, the reference should be added for listed criteria. Anyway, in criteria b) for nitrogen limitation instead of DSi:DIP ratio >16 should stand < 16, if it was presumed that DSi and DIN appear in similar concentrations, though not always the case.

➜We revised them in line 335-338, page 15 based on Dortch and Whitledge (1992).

References: Listed but not cited in the text: Adolf and Harding, 2006; Choi and Noh, 1998; De Oliveira et al 1999; Julian and David, 1966; Kim et al., 2016 Cited in the text but not listed: Choi et al., 1998; Kim et al., 2014; Kwon et al, 2001; Marsh and Weinstein 1966; Paerl et al., 2006; Yun et al., 2014 Cited or listed with different year of publication: Pirt 1975 (cited in the text), listed in references as Pirt 1976 Some references are written in uppercase. To the references published in the same year a, b should be added

➜We revised the references.

In Tables 1 and 4 in April 2012 appears st. 1 which is not marked on the map (Fig.1)

➜We revised the map in Fig. 1.
General Comments The manuscript presents the seasonal variation of biochemical composition of POM in Bay. The author shows the major controlling factor for them based on statistical analysis. Overall, I found the paper to be sound and believe that it contains valuable data in understanding the characteristics of POM and their contribution to coastal ecosystem as basic food source. I think that the paper is worthy of publication for BGS after minor revisions are made, while there are a few areas that need improvement.

Major comment and corrections
1. Page 12, Line 258-278: The author showed _13C value and carbon to nitrogen ratio in surface, in order to find the origin of POM. I think that the contribution of benthic microalgae to POM could be large and significant, since the study area is located in coastal area and extremely turbid condition related to freshwater input or tidal cycles or wind. Therefore, many amounts of benthic microalgae could be included to POM through the resuspension, especially during high river input. Indeed, Table 3 shows the lower 13C value in August.
➔We discussed on potential contributions of benthic microalgae on POM in line 300-302, page 13.

2. Pages 13-14, Line 301-304: For the criteria of their moral ratios among dissolved inorganic nutrients, I wonder could it be applied in coastal area. I think that the status of nutrient limitation in phytoplankton could be different between open oceans and coastal area.
➔Actually, the criteria of the moral ratios can be applied in coastal area based on several papers as we referred in our discussion (e.g., Roelke et al., 1999).

3. Page 15, Line 335-344: As the author discussed, I think that the composition of phytoplankton assemblages and species could be closely related to seasonal variation of biochemical composition. High nitrogen supply during river-input increased season could lead to different phytoplankton composition. For example, the large sized phytoplankton (such as diatom) could be thrived in that condition, since the large phytoplankton could grow best and dominate under eutrophic condition. According to Fernandez et al. (1994), the carbon allocation into different biochemical pools were different depending on dominant phytoplankton group. For example, the carbon allocation into lipids was higher under the dominance of flagellates, whereas the lower lipid synthesis was observed in the dominance of diatoms. Therefore, the seasonally different phytoplankton composition related to nutrient input could affect to the different biochemical composition in the region.
➔Yes, the seasonal compositions of phytoplankton could lead different biochemical compositions. We discussed on that issues in line 380-389, page 15-16.

4. In figure 3, the author shows positive relationship between river input and protein composition. However, I didn't find the positive relationship between them, based on comparison with table 2 and figure 2. For example, the protein composition in August was lowest, although the rive input was considerably high. In addition, the protein composition from October in 2012 to April in 2013 was higher than that in August, even though the lower river inputs were recorded.

➔Actually, the river input data in Table 2 are monthly integrated river inputs to show monthly
patterns of river input and rainfall. In Figure 3, river-inputs were integrated from 20 days prior to our
sampling dates since phytoplankton productivity is recovered after 20 days after rainfall in
Gwangyang Bay according to Min et al. (2011). We added this explanation in line 100-102, page 5 to
make it clear.
Minor corrections
1. Pages 8-9, Line 175-186: The position of some sentences needs to be corrected. For example, the
results about irradiance and chl-a are shown in Table 1 (it is explained in line 178-186). The results
for rainfall and river-input are indicated in former position (in line 175-178), although they are shown
in Table 2.
➔We rearranged the sentences in line 190-194, page 9.
2. Page 9, Line 195-197: The author found that there were no significant differences in spatial
distribution of POM. However, the protein composition in station 2A (is closest to the River) might be
higher than in station 4 and 5, since there is the large effect of river-input on the biochemical
composition in this study.
➔We did ANOVA test for each depth from 3 stations based on an assumption of no spatial difference
and another ANOVA test for a spatial difference by pooling of 3 light depths at one station and
comparing each station based on an assumption of no difference in light depths. But, we found to
realize that there are statistical errors by doing that. So, we deleted no significant differences between
vertical and spatial distributions in our text. The station 2A (is closest to the River) might be the
largest effect of river-input but different effects of river-input could be different depending on water
circulation, tidal currents, winds, and etc as well as distance from the river in Gwangyang Bay. At this
point, we can not determine how much effect at each station from river inputs but the station 2A could
have more proteins than others if they have more influence from river inputs based on Table 1 and Fig.
3.
The manuscript, "The effects of different environmental factors on biochemical composition of particulate organic matter in Gwangyang Bay, South Korea" written by Lee et al., describes the seasonal changes of the biochemical composition such as proteins, lipids, and carbohydrates of the particulate organic matter and investigates the major environmental controlling factors for the changes of biochemical composition. This paper is very interesting to present seasonal biochemical composition of particulate organic matter responding to various environmental factors and the food quantity assessed in Gwangyang Bay since previous studies have focused mainly on different biochemical compositions as the authors mentioned that. The present study has scientific merits and originality in that: 1. the topic, "biochemical composition of particulate organic matter as indicators of food quality and quantity", is very intriguing enough to draw much attention for understanding marine ecosystem especially here in Gwanyang Bay; 2. the study is one of few studies that employed seasonal biochemical composition of particulate organic matter responding to various environmental factors; 3. authors found that river-derived dissolved inorganic nitrogen loading is a major governing factor for determining the biochemical composition in a natural bay system which implies that man-made artificial dams could cause a serious disturbance in the ecosystem. Overall, I would recommend publication of this manuscript for Biogeosciences after some minor revisions. I hope to see authors undertake revisions in an appropriate manner because I really want to see the final version of this paper in print.

Some minor comments are listed below:
-. Be consistent with Gwangyang Bay or bay throughout the text.
➔We checked it throughout the text.
-. Be consistent with average or mean in the results.
➔ We checked it throughout the text.
-. (Section 2.1) What time did usually authors make samplings? I'm wondering if there is tidal influence and/or diurnal changes?
➔ There might be tidal influence (no diurnal changes) so that we sampled waters at high tide period before the noon to reduce the tidal effects. We added sampling times in line 88-89, page 4-5.
-. (line 76) indicate t (ton ?).
➔Yes, it is ton! We changed t into ton in line 78, page 4.
-. (lines 83-83) Describe hot to determine 3 light intensities from secchi disk
➔We described how to determine 3 light depths from secchi disk in line 89-91, page 5.
-. (line 157 in section2.5) Describe what Winberg (1971) is.
➔We described the Winberg equation in line 173, page 8.
-. (3. Results) Indicate statistics results in the result section.
➔We indicated statistics results.
-. (Section 4.1) The first sentence in 4.1 section is repeated as the result section of 3.1. Remove it.
➔We rephrased it.
-. (Table 6) No description for Table 6 is shown in the result section. Describe it in the result section before the discussion on that in the discussion section.

➜We changed Table 6 into Table 2 and described the result for the table.
-. (Section 4.4) The first sentence in 4.4 belongs to the result.
➜We removed the sentence.
-. (Table 7) Be consistent with the unit in Table 7 and the text.
➜We changed the unit in Table 7 in consistency of the text.
-. The conclusion section is needed to be revised since some parts in the conclusion repeat some
results.
➜We removed the conclusion section because it is too general and does not contain the conclusion of
the paper.

[revised manuscript text omitted]